# Neural View Synthesis and Matching for Semi-Supervised Few-Shot Learning of 3D Pose

**Angtian Wang**    **Shenxiao Mei**    **Alan Yuille**    **Adam Kortylewski**

Johns Hopkins University
Maryland, MD 21218, USA
{angtianwang, smei5, ayuille1, akortyl1}@jhu.edu

## Abstract

We study the problem of learning to estimate the 3D object pose from a few labelled examples and a collection of unlabelled data. Our main contribution is a learning framework, neural view synthesis and matching, that can transfer the 3D pose annotation from the labelled to unlabelled images reliably, despite unseen 3D views and nuisance variations such as the object shape, texture, illumination or scene context. In our approach, objects are represented as 3D cuboid meshes composed of feature vectors at each mesh vertex. The model is initialized from a few labelled images and is subsequently used to synthesize feature representations of unseen 3D views. The synthesized views are matched with the feature representations of unlabelled images to generate pseudo-labels of the 3D pose. The pseudo-labelled data is, in turn, used to train the feature extractor such that the features at each mesh vertex are more invariant across varying 3D views of the object. Our model is trained in an EM-type manner alternating between increasing the 3D pose invariance of the feature extractor and annotating unlabelled data through neural view synthesis and matching. We demonstrate the effectiveness of the proposed semi-supervised learning framework for 3D pose estimation on the PASCAL3D+ and KITTI datasets. We find that our approach outperforms all baselines by a wide margin, particularly in an extreme few-shot setting where only 7 annotated images are given. Remarkably, we observe that our model also achieves an exceptional robustness in out-of-distribution scenarios that involve partial occlusion. The code is available at https://github.com/Angtian/NeuralVS.

## 1   Introduction

Object pose estimation is a fundamentally important task in computer vision with a multitude of real-world applications, e.g. in self-driving cars or augmented reality applications. Current deep learning approaches to 3D pose estimation achieve a high performance, but they require large amounts of annotated data to be trained successfully. However, the human annotation of an object's 3D pose is difficult and time consuming, therefore it is desirable to develop methods for learning 3D pose estimation from as few labelled examples as possible.

A powerful approach for training models without requiring a large amount of labels is semi-supervised learning (SSL). SSL mitigates the requirement for labeled data by providing a means of leveraging unlabeled data. Since unlabeled data can often be obtained with low human labor, any performance boost conferred by SSL often comes with low cost. This has led to a multitude of SSL methods, for example for image classification [18, 44], object detection [40] and keypoint localization [31]. However, only limited attention was devoted to SSL for 3D pose estimation.

35th Conference on Neural Information Processing Systems (NeurIPS 2021).

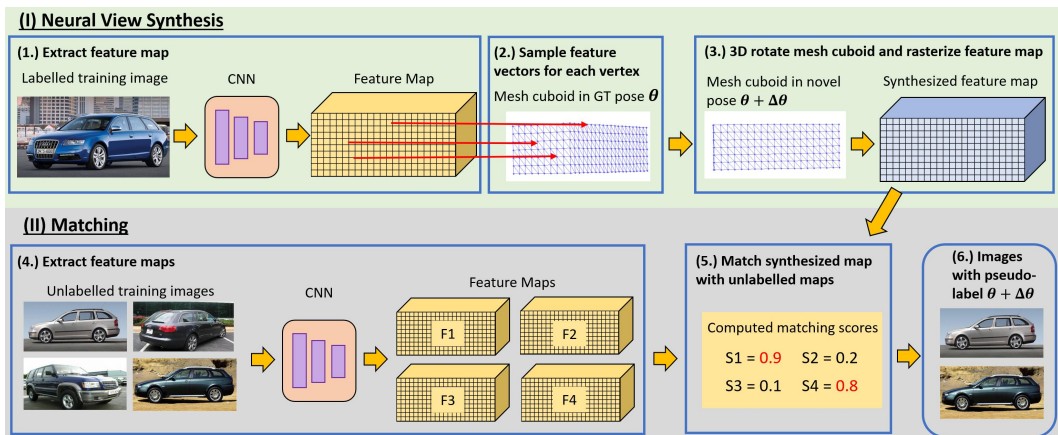

Figure 1: Illustration of how we transfer the 3D pose annotation from one training image to a set of unlabelled images. A detailed description of this process is given in the introduction section.

In this work, we introduce a semi-supervised learning framework for category-level 3D pose estimation from very few annotated examples and a collection on unlabelled images. Intuitively, our framework follows a spatial matching approach, in which we transfer the 3D pose annotation from the labelled examples to the unlabelled data. In general, matching-based approaches aim at transferring annotations between training examples by estimating correspondences between the respective images [29, 2, 26, 15, 8, 45, 19, 27]. However, those prior works mainly focused on transferring 2D annotations, such as segmentation, part annotations, or keypoints, and mostly assume that the objects in the spatially matched images have a similar 3D pose. In this work, we explore the spatial matching of images with objects that have a largely varying 3D pose, in addition to nuisance variations in their shape, color, texture, context and illumination.

**3D pose transfer through Neural View Synthesis and Matching (NVSM).** The intuition behind our approach is illustrated in Figure 1 using a single annotated training image (but note that in practice we use several images). Our method proceeds in two steps: (I) Neural view synthesis and (II) Matching. During neural view synthesis, we start with (1.) extracting the feature map of a training image with a convolutional backbone (CNN) that was pre-trained for image classification. (2.) A mesh cuboid (purple mesh) is projected onto the extracted feature map using the ground-truth 3D pose annotation $\theta$ to sample the corresponding feature vectors at each visible mesh vertex (indicated by red arrows). (3.) The mesh cuboid is subsequently rotated into a novel pose $\theta + \Delta\theta$. By rasterising the sampled feature vectors at the mesh vertices we synthesize a feature map of the object in a novel pose. Subsequently, (4.) we compute the feature maps of the unlabelled training images with the CNN. (5.) We spatially match the synthesized feature map those feature maps of the unlabelled data, resulting in a set of matching scores. (6.) Finally, we assign the 3D pose $\theta + \Delta\theta$ as pseudo-label to those images with highest matching scores (marked in red).

As prior works have shown [41, 58, 5], the features in pre-trained convolutional networks are surprisingly reliable in spatial matching tasks as they are invariant to small variations in nuisance variables such as shape deformations, or changes in the object texture and illumination. This invariance property enables the spatial matching of annotated training images with unlabelled data across varying 3D poses. However, as we demonstrate in our experiments (Section 4.3), the features of the classification pre-trained CNN are not invariant to large 3D pose variations. Therefore, this pseudo-labelling process is initially only accurate for those objects in the unlabelled data that have a similar pose as the object in the annotated training image. This raises the need for improving the 3D pose invariance in the CNN features, to be able to annotate images with larger pose variability.

**Increasing 3D pose invariance in the feature extractor.** We aim to increase the 3D pose invariance in the CNN using the pseudo-labelled data obtained from the NVSM process. The pseudo-labels enable us to extract the features vectors at corresponding mesh vertices in the data. To increase the 3D pose invariance in the CNN, we use a contrastive loss that encourages the feature vectors at a particular mesh vertex to become more similar to each other, while at the same time making them different from the features of other mesh vertices. This contrastive training improves the feature

representation by making it more invariant to changes in the 3D pose, as well as to category-specific nuisance variables such as the object's color, shape, illumination, while also reducing the ambiguity between nearby feature vectors in the feature map, which benefits the spatial matching quality.

The improved CNN features enable us to increase the 3D pose scope $\Delta\theta$ in the NVSM process, and hence to increase the 3D pose variability in the pseudo-labelled data. The pose diversity in the labelled data, in turn, enables us to improve the 3D pose invariance in the CNN feature extractor. We proceed to iterate between pseudo-labelling training data and training the CNN feature extractor, while continuously enlarging the 3D pose variation $\Delta\theta$ of the synthesized views in NVSM. After each update of the CNN, we also update the feature representations at the mesh vertices by computing the moving average of the corresponding feature vectors in the pseudo-labelled images. In this way, the feature representation on the mesh cuboid is continuously adapting to the trained feature extractor.

After the training, the trained CNN and mesh cuboid are used for 3D pose estimation. Given a test image, we first compute the feature map using the CNN and subsequently optimize the 3D pose of the mesh cuboid such that the distance between the features of the projected mesh vertices and the feature map are minimized. We evaluate our model at 3D pose estimation on the Pascal3D+ [53] and KITTI [13] datasets. Our approach proves highly effective in leveraging unlabeled data outperforming all baselines by a wide margin, particularly in an extreme few-shot setting where only 7 or 20 annotated images are given. Remarkably, we observe that our model also achieves an exceptional robustness in out-of-distribution scenarios that involve partial occlusion.

## 2 Related Work

**Object pose estimation.** Object pose estimation is an important and well studied computer vision task. [47] and [32] formulate the pose estimation problem as single step classification problem. In contrast, [25, 35, 59] solve the object pose estimation problem via a two-step approach, which involves keypoint detection and solving a Perspective-n-Point (PnP) process. Recently, [51, 56] demonstrate the success of analysis-by-synthesis approaches for object pose estimation, which use a differentiable renderer to generate a synthesised image and estimate the object pose by minimizing a reconstruction loss. [49] extend the render-and-compare approach for pose estimation to the neural feature level, which significantly decreases the difficulty of the optimization process during pose estimation, while also improving the robustness. However, all of these approaches require a large amount of data with annotated 3D object pose during training, which is time consuming and expensive. In this work, we follow a matching-based approach, in which we transfer the 3D annotation form a few labelled images to a collection of unlabelled data, thus leading to a largely enhanced data efficiency.

**Spatial Matching.** Spatial image matching aims at estimating the correspondence between objects in two different images. Traditional matching algorithms use features from corners [30, 16], blobs [28], edges or lines [9, 43]. Recent works [42, 24, 4] demonstrate the advantage of utilizing deep neural network features for image matching. [52, 6] leverage the 3D structure of objects as additional constraint in the spatial matching process, for objects in similar poses. [2] demonstrate that neural feature matching can help reducing the amount of training data required for semantic part detection. In this work, we spatially match objects with largely varying 3D pose and leverage this ability for the semi-supervised few-shot learning of a model for 3D pose estimation.

**Semi-supervised Learning.** Semi-supervised learning [10, 60, 48] aims at reducing the amount of annotations by utilizing unlabelled data. Self-training [55, 46, 33] is one of the most widely used semi-supervised learning approaches to create pseudo-labels, which are in turn used as supervision during the learning process. Current works show the success of self-training in classification[18, 44], detection[40], and keypoint localization[31]. However, self-training for object pose estimation has not been well explored yet. In this work, we propose a semi-supervised pose estimation approach, that can be trained using very few annotations, while also being highly robust to partial occlusion.

**Robust Vision through Approximate Analysis-by-Synthesis.** In a broader context, our work builds on and extends a recent line of work that follows an *approximate analysis-by-synthesis* approach to computer vision [49], which formulates vision as an inverse rendering process on the level of neural network features. Several recent works demonstrate that approximate analysis-by-synthesis induces a largely enhanced generalization in out-of-distribution situations such as when objects are partially occluded in image classification [21–23, 57] and object detection [50], when images are modified

through adversarial patches [20], or when objects are viewed from unseen 3D poses [49]. Our work enables the learning of models for approximate analysis-by-synthesis with minimal supervision.

## 3 Method

In this section, we describe our approach for the semi-supervised learning of 3D pose from a few labelled examples and a collection of unlabelled data. The central part of our framework is a novel method for generating pseudo annotations of the 3D pose in unlabelled images through synthesizing novel views of an object on the level of neural network features (Section 3.1) and matching those synthesized views to the features of unlabelled images (Section 3.2). We discuss how this method can be integrated into a semi-supervised learning pipeline for 3D pose estimation in Section 3.3.

### 3.1 Neural View Synthesis

A key requirement for an efficient semi-supervised learning is the ability to generate reliable pseudo labels for the unlabelled data using only few labelled images. The challenge when generating pseudo-labels of an objects 3D pose is that it requires generalizing to situations where an object is depicted in a previously unseen 3D pose, which has proven to be a challenging problem for discriminative models [1, 37]. In this work, we generate pseudo-labels by synthesizing novel views of an object on the level of neural network feature activations. In the following, we explain the details of this process based on a single labelled training image, and we discuss later how this is extended to multiple images.

**Feature extraction.** Our approach starts with an image $I$ and the corresponding 3D pose annotation $\theta \in \mathbb{R}^3$. We use a convolutional neural network $\Psi$ to extract a feature map $F = \Psi(I) \in \mathbb{R}^{H \times W \times C}$ with $C$ being the number of channels. To start with, we use an ImageNet [12] pre-trained neural network as feature extractor and we discuss how the feature extractor is trained in Section 3.3.

**Mesh cuboid and feature sampling.** We aim to synthesize the object in the feature map $F$ from a novel view. To achieve this, we represent the object as a 3D cuboid mesh composed of a set of vertices $\Gamma = \{x_r \in \mathbb{R}^3 | r = 1, \ldots, R\}$ and feature vectors at each mesh vertex $\Sigma = \{\sigma_r \in \mathbb{R}^D | r = 1, \ldots, R\}$. The mesh cuboid serves as 3D representation of the object in the feature map and allows us to change the 3D pose of the object as described in the following. Given an anchor image $I$ and the corresponding feature map $F$, we sample the mesh features $\Sigma$ from the feature map using the ground truth pose annotation $\theta$. For this, we project the mesh vertices $\Gamma$ into the feature map by multiplying the vertices with a weak perspective projection matrix $P_\theta$:

$$\sigma_r = F(P_\theta \cdot x_r). \tag{1}$$

where $\sigma_r$ is the sampled feature vector at the position of the $r$-th projected mesh vertex. Note that the mesh cuboid does not represent the object shape in detail and therefore also feature vectors at the background will be sampled. But intuitively the model will learn that some features are coherent across views and therefore will be able to focus more on the foreground region.

**Synthesizing feature maps in novel views with rasterisation.** Given the mesh cuboid and the sampled feature vectors, we can synthesize novel views of the object, by rotating the mesh cuboid into a novel 3D pose $\theta' = \theta + \Delta\theta$. Subsequently, we generate a feature map of the object in this novel view through rasterisation:

$$F_{\theta'} = \Re(\Gamma, \Sigma, \theta') \in \mathbb{R}^{H \times W \times C}, \tag{2}$$

where $\Re$ denotes the rasterisation process of the cuboid mesh $\Gamma$ with the vertex features $\Sigma$ in the 3D pose $\theta'$. Note that rasterisation is a standard technique in computer graphics to draw 3D models into 2D images [38]. However, instead of drawing images with RGB pixels, in our usecase we draw feature maps. The rasterisation resolves two problems. First, when projecting a 3D mesh into an image, some of the vertices will become occluded due to self-occlusion, and rasterisation accounts for this effect. Second, some pixels in an image might not be covered by a projected vertex and to fill these holes in the image, rasterisation interpolates between the features of the mesh vertices.

### 3.2 Spatially Matching Synthesized Views to Unlabelled Images

We generate pseudo-labels of an object's 3D pose by spatially matching synthesized object views $F_{\theta'}$ to the set of unlabelled images $\mathbb{U} = \{I_1, \ldots, I_M\}$. During matching, we compute the similarity

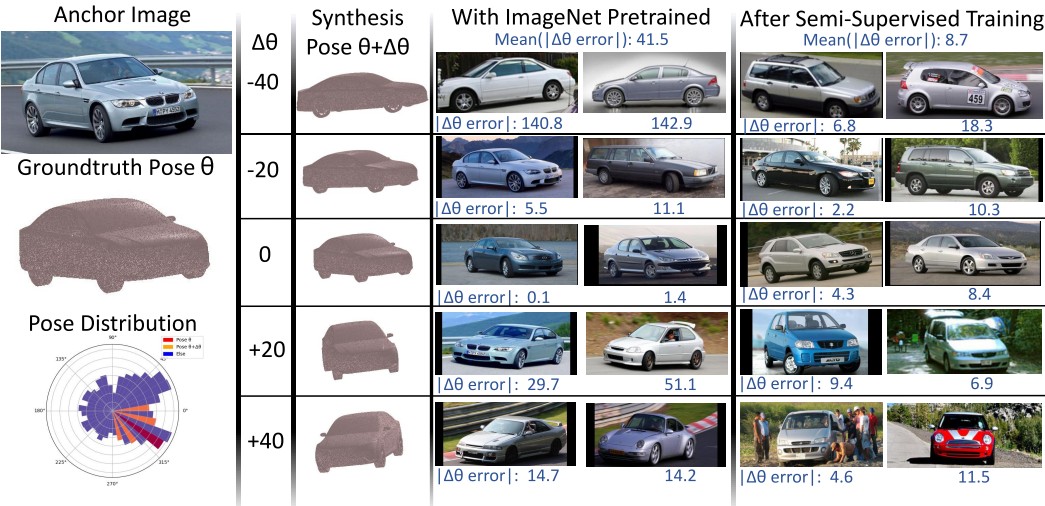

Figure 2: Spatial matching results of synthesized views. We synthesize novel views of the anchor image using the ground-truth pose $\theta$ and varying azimuth angles $\theta + \Delta\theta$ (here we sample $\Delta\theta$ along azimuth). We spatially match the synthesized views with all images in the data and retrieve those with highest matching scores. We also report the pose difference of the retrieved images. The car CAD model is only used for visualization purposes and is not used in our method. Note how the retrieval results are much more accurate after our proposed semi-supervised training, compared to when using pre-trained features.

between $F_{\theta'}$ and the feature map of an unlabelled image $F_m = \Psi(I_m)$ as the sum of the cosine distances $d(\cdot, \cdot)$ between their corresponding feature vectors:

$$S(F_{\theta'}, F_m) = \frac{1}{HW} \sum_h \sum_w [1 - d(F_{\theta'}(h, w), F_m(h, w))]. \tag{3}$$

We retrieve those images with highest spatial matching similarities that greater than a pre-defined threshold $\tau$, and pseudo-label them with the 3D pose of the mesh cuboid $\theta'$.

**The role of pose invariance on the spatial matching quality.** The main challenge when spatially matching a synthesized view $F_{\theta + \Delta\theta}$ to a set of unlabelled images is to achieve a reliable spatial matching result when $\Delta\theta$ is large, and hence when the novel view is very different from the original 3D pose of the object in the labelled training image. Figure 2 illustrates this problem. In our work, we start with an ImageNet pre-trained CNN as feature extractor $\Psi$. We observe that these features have some invariance to 3D pose and other nuisances, therefore the retrieved images are consistent with the pseudo-labelled pose when $\Delta\theta$ is relatively small. However, with an increased value of $\Delta\theta$ the objects in the retrieved images are inconsistent with the 3D pose of the mesh cuboid (refer to section 4.3 for more details). Therefore, we keep $\Delta\theta$ small in the initial neural view synthesis and matching process. This allows us to collect additional data with reliable 3D pose annotation, although with limited pose variability. To increase the pseudo annotation to larger unseen pose angles, we subsequently train the feature extractor to become more pose invariant, using the pseudo-labelled data as described in the following section.

### 3.3 Semi-Supervised Learning of 3D pose

Neural view synthesis (Section 3.1) and matching (Section 3.2) process enables us to annotate the 3D pose in unlabelled data, but this requires features that are invariant to 3D pose. However, the training of a feature extractor with higher 3D pose invariance requires amount of annotated data. To resolve this chicken-and-egg problem, we iterate between annotating data and training the feature extractor in an Expectation-Maximization-type manner.

**Training the feature extractor.** After a first pass of neural view synthesis and matching with an ImageNet pre-trained feature extractor $\Psi$, we obtain a set of annotated data $\mathbb{P} = \{(I_i, \theta_i) | i = 1, \dots, N\}$ that is, in turn, used to improve the invariance to viewpoint changes in the feature extractor with a contrastive training strategy [3, 11]. Specifically, given data $\mathbb{P}$ and the mesh cuboid $\Gamma$, we use

a contrastive loss to maximize the feature similarity between the feature vectors that correspond to the same vertex in different images:

$$L_+(F_i, F_j, \Gamma) = \sum_{r=1}^{R}[1 - d(F_i(P_{\theta_i} \cdot x_r), F_j(P_{\theta_j} \cdot x_r))]. \tag{4}$$

Here $F_i = \Psi(I_i)$ and $F_j = \Psi(I_j)$ are feature maps of two images with pose annotations $\theta_i$ and $\theta_j$. At the same time, we minimize the similarity between feature vectors of different vertices:

$$L_-(F_i, F_j, \Gamma) = \sum_{r=1}^{R} \sum_{r' \neq r} d(F_i(P_{\theta_i} \cdot x_r), F_j(P_{\theta_j} \cdot x_{r'}))]. \tag{5}$$

During training, the overall loss is computed over all combinations of images in $\mathbb{P}$. After training the feature extractor, we can resort to an improved pseudo-labelling through neural view synthesis and matching, which will in turn enable us to improve the feature extractor.

Throughout the EM-type learning process, we gradually increase the pose difference $\Delta\theta$ of the synthesized views from the annotated training data. Figure 1 illustrates the improved retrieval performance after several EM-type iterations of NVSM and training the feature extractor using only the single depicted training image.

**Semi-supervised learning with multiple annotated images.** While our approach is very data efficient, we observe that a single training example is not sufficient to bootstrap the full learning process successfully, because objects in images not only vary due to their 3D pose, but also in terms of nuisance factors such as their texture, shape, context or illumination conditions. To generalize our approach to multiple images, we initialize the mesh features $\Sigma$ by averaging over the whole set of annotated training images $\mathbb{A} = \{(I_a, \theta_a) | a = 1, \ldots, A\}$:

$$\sigma_r = \sum_a F_a(P_{\theta_a} \cdot x_r). \tag{6}$$

The training of the feature extractor naturally extends to a training with multiple annotated images, as the only thing that changes is the size of the pseudo-annotated dataset $\mathbb{P}$. However, note that after updating the feature extractor, the mesh features $\Sigma$ also need to be updated. After the first training iteration, we leverage the pseudo-labelled dataset $\mathbb{P}$ to compute the mesh features $\Sigma$, since it is multiple times larger compared to the annotated dataset $\mathbb{A}$. To compute the update at a reasonable computational cost, we use a moving average update [3]:

$$\sigma_r^{t+1} = (1 - \alpha) * \phi_r^t + \alpha * \sum_n F_n(P_{\theta_n} \cdot x_r), \tag{7}$$

where $\sigma_r^t$ is the current mesh feature at time step $t$ and $\sigma_r^{t+1}$ is the updated feature. The moving average update enables us to update the mesh features while training the feature extractor, and hence without requiring to process the whole set $\mathbb{P}$ again, after training the feature extractor. In practice, we observe that this improves the runtime of the training, while neither hurting nor benefiting the overall performance significantly.

**3D pose estimation with the mesh cuboid.** After training the model, we can leverage the feature extractor and the mesh cuboid directly to estimate the 3D pose of an object. Following related works on analysis-by-synthesis for face analysis [7] or pose estimation with neural network features [49], we implement 3D pose estimation in a render-and-compare manner. Specifically, given a test image $I_T$ we first compute its feature representation using the trained backbone $F_T = \Psi(I_T)$. Subsequently, we optimize the 3D pose $\theta$ of the mesh cuboid $\Gamma$ with learned mesh features $\Sigma$, such that the synthesized feature map $F_\theta$ most closely resembles $F_T$, i.e. such that $S(F_T, F_\theta)$ is minimized.

## 4 Experiments

We describe the experimental setup in Section 4.1. Subsequently, we study the performance of our approach at semi-supervised few-shot learning of 3D pose estimation in Section 4.2. We diagnose the how the spatial matching quality evolves during training in Section 4.3.

## 4.1 Experiment Setup

**Evaluation.** The task of 3D object pose estimation involves the prediction of three rotation parameters, i.e. azimuth, elevation, in-plane rotation, of an object relative to the camera. We follow the evaluation protocol used in previous works [59, 49] and assume the object scale and center are given, and measure the pose estimation error between the predicted rotation matrix and ground-turth rotation matrix: $\Delta\left(R_{pred}, R_{gt}\right) = \frac{\left\|\log m\left(R_{pred}^T R_{gt}\right)\right\|_F}{\sqrt{2}}$. We report accuracy w.r.t. two thresholds $\frac{\pi}{6}$ and $\frac{\pi}{18}$, as well as, the median error of the prediction.

**Dataset.** We report 3D pose estimation performance for all methods on the PASCAL3D+ dataset [54] and the KITTI dataset [14]. For Pascal3D+, we evaluate 6 vehicle categories (aeroplane, bicycle, boat, bus, car, motorbike), which have a relatively evenly distributed pose regarding the azimuth angle. We evaluate few-shot semi-supervised learning in 3 few-shot settings which train on 7, 20, and 50 annotated images for each category respectively. The 7 annotated images are selected such that they are spread around the pose space (see supplementary for details) and the remaining images are sampled randomly. For the KITTI dataset, we utilize the 3D detection annotation of the car category to crop the objects such that they are located in the image center and to compute the 3D object pose. Following the official KITTI protocol, we split the dataset such that it contains 2047 training images and 681 testing images, including 411 fully visible 201 partially occluded and 69 largely occluded images according to the occlusion level annotation.

**Training Setup.** We use an ImageNet [12] pre-trained ResNet50 [17] as feature extractor. The dimensions of the cuboid mesh $\Gamma$ are defined such that for each category most of the object area is covered. We sample $\Delta\theta$ at fixed distances of 10 degree during neural view synthesis. During training we continually increase the pose range by 10 degrees after each iteration of training the feature extractor until the whole pose space is covered. For each sampling step, we pseudo label between 20 to 50 images depending on the category and the similarity threshold $\tau = 0.9$. In total, we train the whole model by alternating 100 times between neural view synthesis and matching (NVSM) and training the feature extractor, which takes around 5 hours per category on a machine with 4 RTX Titan GPUs. We implement our approach in PyTorch [34] and apply the rasterisation implemented in PyTorch3D [39].

**Baselines.** We evaluate StarMap [59] as our baseline as it is one of the state-of-the-art supervised approaches to 3D pose estimation. Moreover, we evaluate NeMo [49] which is a recently proposed approach for 3D pose estimation that is robust and data efficient. For NeMo, we use the same single mesh cuboid as our method is using. We also implement a baseline that formulates the object pose estimation problem as a classification task as described in [59]. For the classification approach, we evaluate both a category specific (Res50-Spec) and a non-specific (Res50-Gene) ResNet50 classifier. The former formulates the pose estimation task for all categories as one single classification task, whereas the latter learns one classifier per category.

Note that all baselines are evaluated using a semi-supervised few-shot protocol. This means we use 7, 20, and 50 annotated images for training. In order to utilize the unlabelled images, we use a common pseudo-labelling strategy for all baselines. Specifically, we first train a model on the annotated images, and use the trained model to predict a pseudo-label for all unlabelled images in the training set. We keep those pseudo-labels with a confidence threshold $\tau = 0.9$, and we utilize the pseudo-labeled data as well as the annotated data to train the final model.

## 4.2 Semi-Supervised Few-Shot 3D Pose Estimation

**Pascal3D+.** Table 1 shows the performance ouf our approach and all baselines at semi-supervised few-shot 3D pose estimation on the PASCAL3D+ dataset. All models are evaluated using 7, 20, and 50 (per class) training images with annotated 3D pose and a collection of unlabelled training data (as described in Section 4.1). The Resnet50 classification baselines achieves a reasonable performance. Notably, the category-specific classifier performs significantly lower compared to the classifier trained for all categories, suggesting that the latter benefits from the additional data of all categories. In particular, for the extreme low data setting using only 7 annotated examples, the model can share information across categories which greatly improves the accuracy by 5.5%. Remarkably, StarMap performs worse compared to the simple classification baseline. This highlights the relevance the semi-supervised few-shot setting as it shows that current state-of-the-art models for 3D pose

Table 1: Few-shot pose estimation results on PASCAL3D+. We indicate the number of annotations during training for each category and evaluate all approaches using Accuracy (in percent, higher better) and Median Error (in degree, lower better). We also include the fully supervised baseline [49] (Full Sup.) which is trained from hundreds of images per category.

| Metric | $ACC_{\frac{\pi}{6}} \uparrow$ | | | | $ACC_{\frac{\pi}{18}} \uparrow$ | | | | $MedErr \downarrow$ | | | |
|---|---|---|---|---|---|---|---|---|---|---|---|---|
| Num Annos | 7 | 20 | 50 | Mean | 7 | 20 | 50 | Mean | 7 | 20 | 50 | Mean |
| Res50-Gene | 36.1 | 45.2 | 54.6 | 45.3 | 14.7 | 25.5 | 34.2 | 24.8 | 39.1 | 26.3 | **20.2** | 28.5 |
| Res50-Spec | 29.6 | 42.8 | 50.4 | 40.9 | 13.3 | 23.0 | 29.3 | 21.9 | 46.5 | 29.4 | 23.0 | 32.9 |
| StarMap | 30.7 | 35.6 | 53.8 | 40.0 | 4.3 | 7.2 | 19.0 | 10.1 | 49.6 | 46.4 | 27.9 | 41.3 |
| NeMo | 38.4 | 51.7 | **69.3** | 53.1 | 17.8 | 31.9 | **45.7** | 31.8 | 60.0 | 33.3 | 22.1 | 38.5 |
| Ours | **53.8** | **61.7** | 65.6 | **60.4** | **27.0** | **34.0** | 39.8 | **33.6** | **37.5** | **28.7** | 24.2 | **30.1** |
| Full Sup. [49] | — | — | — | 89.3 | — | — | — | 66.7 | — | — | — | 7.7 |

Table 2: Few-shot pose estimation results on the KITTI dataset at different levels of partial occlusion. We report results using prediction accuracy and median error.

| Eval Metric | Occ level | Fully visible | | | Partially occluded | | | Largely Occluded | | | Mean |
|---|---|---|---|---|---|---|---|---|---|---|---|
| | Num Annos | 7 | 20 | 50 | 7 | 20 | 50 | 7 | 20 | 50 | |
| $ACC_{\frac{\pi}{6}} \uparrow$ | NeMo | 34.3 | 83.9 | 89.8 | 14.9 | 58.2 | 74.6 | 4.3 | 27.5 | 30.4 | 58.5 |
| | Ours | **84.2** | **94.6** | **97.6** | **68.2** | **88.6** | **92.5** | **52.2** | **60.9** | **63.8** | **86.1** |
| $ACC_{\frac{\pi}{18}} \uparrow$ | NeMo | 17.3 | 74.9 | 81.5 | 4.5 | 37.8 | 61.7 | 0.0 | 7.2 | 11.6 | 45.8 |
| | Ours | **23.2** | **81.1** | **88.3** | **18.7** | **81.6** | **82.1** | **16.4** | **39.1** | **42.0** | **60.0** |
| $MedErr \downarrow$ | NeMo | 64.1 | 5.9 | 5.6 | 84.1 | 13.2 | 7.8 | 99.9 | 59.8 | 50.0 | 32.6 |
| | Ours | **20.3** | **8.1** | **4.1** | **24.0** | **12.2** | **5.3** | **27.0** | **13.3** | **12.2** | **12.4** |

estimation are designed for data rich training and cannot simply be extended to semi-supervised training setups. This deficit of generalizing from few annotated data is particularly prominent in the high accuracy evaluation $\frac{\pi}{18}$. In contrast to StarMap, the NeMo baseline shows higher data efficiency. This can be attributed to the generative nature of the model which enables it to become more data efficient compared to the discriminatively trained baselines. However, NeMo still requires at least 50 annotated images per category to perform well, while having a significantly reduced performance in the training settings with fewer data.

Our proposed approach gives significantly higher accuracy compared to all baselines in the low data regime using 7 and 20 annotated images, while being competitive in the higher data regime using 50 annotated examples. Specifically, our model achieves exceptionally high performance using 7 annotated images improving the accuracy by $15.4\%@\frac{\pi}{6}$ and $9.2\%@\frac{\pi}{18}$.

**KITTI and occlusion robustness.** Table 2 shows the results on the KITTI dataset. Using the occlusion annotation in the test data we also study the robustness of the models under occlusion. We compare our approach to the NeMo baseline only, as it is the most competitive model (see original paper [49] for comparisons). Moreover, StarMap cannot be trained on the KITTI dataset, because it requires keypoint annotations, which are not provided in the data. Notably, our approach outperforms NeMo in all experiments by a wide margin. The most prominent performance gain is observed in the extreme few-shot setting of using 7 images only. A notable performance increase can be observed when increasing the annotated data to 20, while more data, i.e. 50 data does not result in a comparable increase. Interestingly, our model is also highly robust to partial occlusion, outperforming NeMo under low and large partial occlusion scenarios. Note that this is an out-of-distribution scenario, since the training data does not contain partially occluded objects. The overall improved performance compared to Table 1 can be attributed to the fact that KITTI only contains 3D annotations of cars, which have a cuboid like overall shape. In contrast, PASCAL3D+ contains other objects, such as aeroplanes, for which the shape is not approximated well by a cuboid. This suggests, that a more accurate shape representation could further improve the performance on the PASCAL3D+ data. Qualitative prediction results of our method are illustrated in Figure 3 for a number of different categories in the PASCAL3D+ and KITTI datasets.

## 4.3 Quality of Neural View Synthesis and Matching over Time

Figure 4 illustrates the quality of the neural view synthesis and matching process with different feature extractors. We start from a set of 20 randomly selected anchor images from the car category of

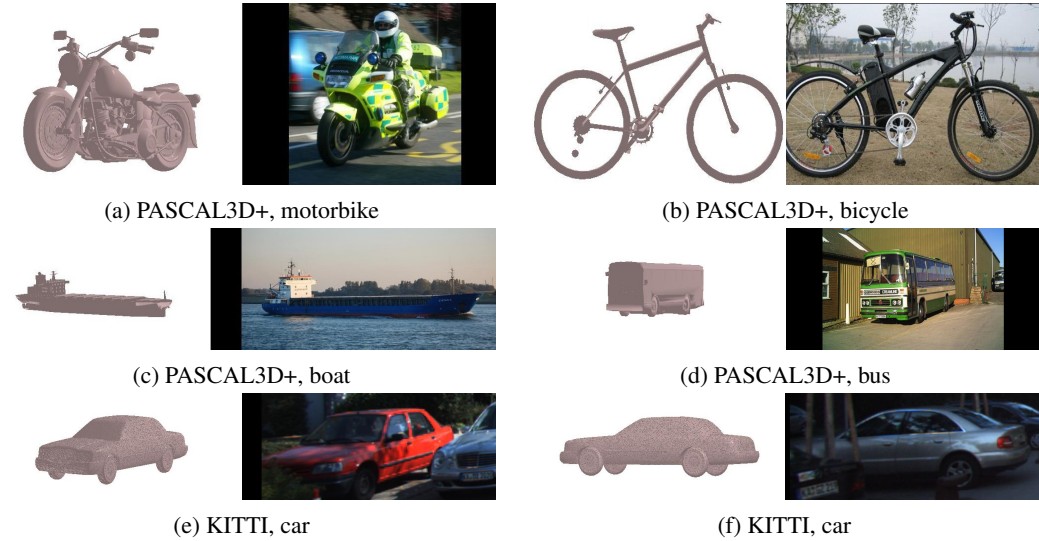

(a) PASCAL3D+, motorbike           (b) PASCAL3D+, bicycle

(c) PASCAL3D+, boat           (d) PASCAL3D+, bus

(e) KITTI, car           (f) KITTI, car

Figure 3: Qualitative results on the PASCAL3D+ and the KITTI dataset. We illustrate the predicted 3D pose using a CAD model. Note that the CAD model is not used for our approach.

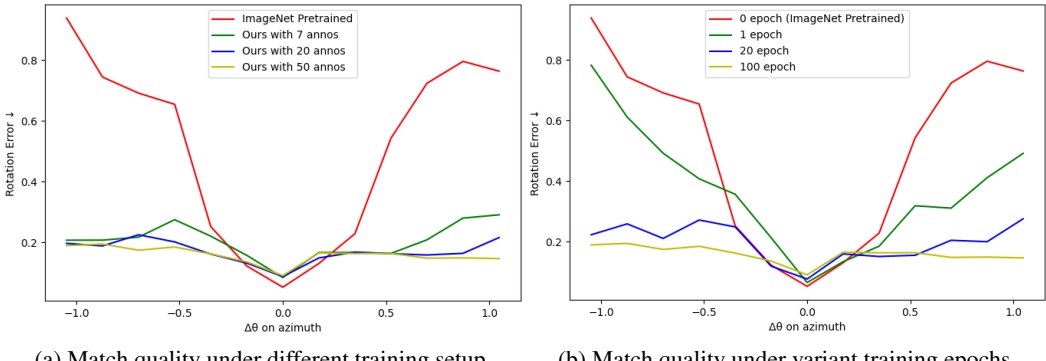

(a) Match quality under different training setup.      (b) Match quality under variant training epochs.

Figure 4: Visualization of the spatial matching quality comparing an ImageNet pre-trained feature extractor and feature extractors trained with NVSM with different amount of annotated data. We plot the 3D pose rotation error of spatially matched images (y-axis) as a function of the target azimuth angle used for synthesizing novel views (x-axis). (a) Influence of the number of annotated images. (b) Match quality at different training epochs for a model trained with 50 annotated images.

the PASCAL3D+ dataset. For each anchor image, we use the ground-truth 3D pose $\theta$ and synthesize novel views as described in Section 3.1 by varying the azimuth angle of the 3D pose (x-axis). We spatially match the synthesized views to the remaining test data as described in Section 3.2 to retrieve 3 images that best fit the synthesized views. The y-axis of each plot shows the rotation error between the cuboid pose $\theta + \Delta\theta$ used to synthesize the novel view and the retrieved images. Each plot is averaged over all anchor images and plots the error as a function of the azimuth pose in the range from $-\frac{\pi}{3}$ to $\frac{\pi}{3}$. Figure 4(a) compares the spatial matching quality of an ImageNet pre-trained feature extractor (red) with feature extractors that are trained with our proposed framework and different numbers of annotated images. It can be observed, that the ImageNet pre-trained features are reasonably effective when the synthesized pose is close to the ground-truth pose, but they are not reliable when the pose difference is large. Remarkably, when using 7 annotated images and NVSM, our model is able to train the feature extractor that is much more reliable even for very large pose differences. We also observe that additional annotated data further improves the spatial matching quality. Figure 4(b) shows how the matching quality evolves as a function of the number of trained epochs using 50 annotated training images. The matching quality improves significantly over the first 20 epochs and further improves, but more slowly, over the remaining training process.

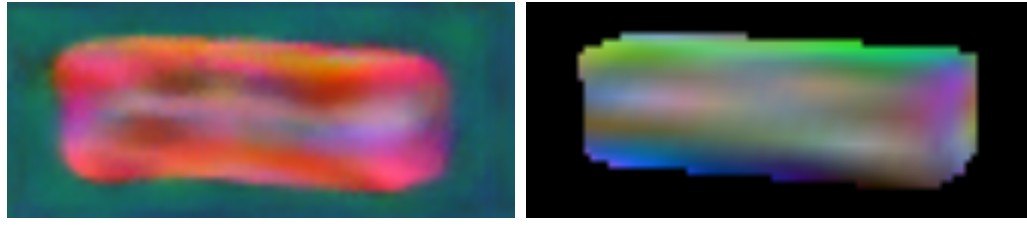

|(a) Extracted Features|(b) Averaged Features|

Figure 5: Visualization for features obtained from semi-supervised trained feature extractor. We use PCA to reduce the features into 3-channels and visualize them in RGB. (a) Features extracted from one testing image via semi-supervised trained feature extractor. (b) The learnt average features of the training set $\sigma_r$.

## 4.4 Visualizing the Learned Features

In order to qualitatively evaluate the features and the feature extractor learned with our semi-supervised approach, we visualize the features using Principal Component Analysis (PCA) [36]. In Figure 5 (a), we apply PCA on the features extracted from a test image, and reduce their dimensionality to three using PCA to visualize them as an RGB image. In Figure 5 (b), we conduct the same dimensionality reduction on the averaged feature vectors (as described in Equation 6) at each mesh vertex $\Sigma$, and render them using the same 3D pose as in Figure 5 (a).

Figure 5 (a) shows a clear boundary between the object and background, which indicates that the ImageNet pre-trained backbone backbone is object-aware and encodes features on the object very differently compared to the background context. However, features within the object are similar to each other. In contrast, Figure 5 (b) demonstrates that the features of our trained features extractor are more similar to each other when they are spatially close. Moreover, most features on the mesh have a different color embedding, i.e. they are different from each other. This is exactly what the contrastive loss in Equations 4 and 5 encourages. However, we note that the first three PCA components only account for about $14.5\%$ of the variation in the data (and hence some features that are very different in the original space might share a similar color in this visualization).

## 5 Conclusion

In this work, we introduced a semi-supervised learning framework for category-level 3D pose estimation from very few annotated examples and a collection on unlabelled images. We followed a matching-based approach, in which we transfer the 3D pose annotation from the labelled examples to the unlabelled data. We achieved this by representing objects as 3D cuboid meshes composed of feature vectors at each mesh vertex. The mesh features were sampled from the annotated data and subsequently used to generate feature maps of unseen 3D views by rotating the mesh cuboid. The synthesized views were then spatially matched to unlabelled data to generate pseudo-labels. We demonstrated that the proposed neural view synthesis and matching approach enables a highly efficient learning of 3D pose, particularly in extreme few shot scenarios, while at the same time achieving an exceptional robustness to out-of-distribution scenarios that involve partial occlusion.

**Limitations and Societal Impact.** Our work shares a common limitation with other works on 3D pose estimation, in that it cannot be trivially generalized to articulated objects. This is even more emphasized in the semi-supervised few-shot setting that we are studying in this paper. For us, the most promising future research direction would therefore be to extend our efficient learning framework to articulated objects such as humans or animals. Like other approaches for 3D pose estimation, our framework belongs to the type of technical tools that do not introduce any additional foreseeable societal problems, but will in the long term make computer vision models generally better.

## Acknowledgments

We gratefully acknowledge funding support from Institute for Assured Autonomy at JHU with Grant IAA 80052272, NIH R01 EY029700, and ONR N00014-20-1-2206. We also thank Qing Liu, Zihao Xiao, Peng Wang and Fang Yan for suggestions on our paper.

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
