# OpenReview forum: "Neural View Synthesis and Matching for Semi-Supervised Few-Shot Learning of 3D Pose"
_NeurIPS.cc/2021/Conference — NeurIPS 2021 Poster_

### Official Review · Reviewer_9Mv2 · 2021-07-16

**Rating:** 8
**Confidence:** 4

**Summary:**

This paper investigates a new problem of semi-supervied 3D viewpoint estimation using few-shot labeled examples. It further proposes a new method for the task based on learning a 3D cuboid feature for a category with features at its vertices which are viewpoint invariant. The authors propose a method of rotating the cuboid to generate feature representations for novel views. Another key insight of the paper is how to expand the ranges of novel views for which the cuboid can be rotated by introducing a contrastive loss to update the cuboid feature representation to be more rotation invariant by combining with the pseudo--labeled images. The authors compare their method to two baseline SoTA methods and outperform the existing methods by a large margin in the few-shot setting.


**Limitations And Societal Impact:**

It would be good to discuss further the points 1 and 2 stated in the previous answer as potential limitations of the method in the paper, if they are indeed limitations of the method.

**Main Review:**

Originality: The paper introduces both a new challenging task of few-shot semi-supervised learning as well as a novel and effective method for it with a strong baseline performance. The work is quite original and opens up a new research area.

Quality: The proposed method is technically sound. The experiments are adequate and well formulated. I note a few shortcoming to improve the paper.

1. The authors describe that they manually chose the 7 few-shot labeled examples per category to train with such that they equally span the entire range of pose angles. It seems to me that this choice is critical to the success of their proposed method. In the worst case, if all the 7 few-shot poses were clustered at a single point, I doubt that the authors would be able to expand the pseudo labels to beyond a small range of angles around the labeled ones. The authors should discuss this limitation of their approach more clearly and provide experiments to clearly show the affect of the choice of few-shot samples on the accuracy of their algorithm. Does their algorithm support a random choice of angles?

2. From Table 1, the proposed method does not seem to perform as well as NeMo for 50 training samples. This behavior for larger number of trained samples is not observed for the KITTI datasets, however. So, perhaps it is an artifact of the Pascal3D+ dataset. I would like to hear a further discussion of this from the authors and whether they have analyzed  the cause of this observation. I am also curious as to what happens with greater than 50 labeled samples and for 100% labeled training data? Does the proposed approach perform as well as NeMo on both datasets in fully labeled scenarios as well or is it limited in applicability only to less labeled scenarios?

3. Few-shot Viewpoint estimation has been explored previously in the work: Few-Shot Viewpoint Estimation, Tseng et al, BMVC 2019. However it was in a different context of exploring few-shot generalization to previously unseen categories of objects, which is different from this current work. Nevertheless it would be good to cite this closely related prior work.

Clarity: The paper is mostly clear. Code is provided. There are several small typos throughout the paper which would be helped with a thorough editorial review.

Significance: Labeling viewpoint or 3D pose is a hard problem. The proposed work opens up a new line of research; proposes a novel method with several interesting new insights, including the use of the 3D feature projection and of using the contrastive loss in an EM-style framework to expand to more viewpoint; performs significantly better than the SoTA approaches and generalized to many different object categories. This is a significant and broadly impactful piece of research.

**Time Spent Reviewing:**

3

---

> ### Author Response · Authors · 2021-08-10
> **Response to Reviewer 9Mv2**
>
> We thank the reviewers for their very helpful feedback. In the following, we address each individual question in detail. We ask the reviewers to please follow-up with us in the discussion period in case of any remaining questions.
>
> **Q1: The authors describe that they manually chose the 7 few-shot labeled examples per category to train with such that they equally span the entire range of pose angles. It seems to me that this choice is critical to the success of their proposed method. In the worst case, if all the 7 few-shot poses were clustered at a single point, I doubt that the authors would be able to expand the pseudo labels to beyond a small range of angles around the labeled ones. The authors should discuss this limitation of their approach more clearly and provide experiments to clearly show the affect of the choice of few-shot samples on the accuracy of their algorithm. Does their algorithm support a random choice of angles?**
>
> Thank you for pointing that we should discuss the importance of the labeled images in our semi-supervised few-shot learning setting. We will add the following discussion on that topic to our paper.
>
> In general, the choice of the few-shot labels is a common challenge for many semi-supervised few-shot learning approaches, for example, in image classification [J, K, L]. Admittedly, our approach is affected by the choice of the labeled images, and in our work, we choose the starting images such that they are widely distributed in the pose space (note that in our experiments, we use the same labeled data all methods, including the baselines ). In the following, we show our model's 3D pose estimation results when trained from 7 randomly sampled labeled images. We discuss in detail why we expect a random sampling to significantly reduce our method's performance (as well as the performance of the baselines).
>
> The following table shows the average performance of two training runs with seven randomly sampled labeled images. Due to time constraints, we run the experiment on three object categories (car, bus, motorbike). We provide the result on these three categories (and if feasible, we will update the result with all  object categories in the discussion, but we do not expect large deviations):
>
> |    Labeled images   | Pi / 6 | Pi / 18 | Med  |
> |---------------------------|---------|----------|--------|
> | Mannually selected | 75.6  | 41.8    | 16.8 |
> | Randomly selected | 52.9  | 22.3    | 32.7 |
>
> We observe a performance decrease of 22.7% in terms of pi/6 and 19.5% in terms of pi/18 accuracy. There are two reasons why this performance decrease is to be expected:
> 1) The pose distribution in the PASCAL3D+ data is highly unbalanced, as shown in the following Figure, where we visualize the distribution of the azimuth angle of the three categories used in the experiment above:
> Link to an anonymous google driver: https://drive.google.com/file/d/1Za2BLzCycBprb1vH8vVdeK_raQ6KlAZE/view?usp=sharing.
> Note how objects in frontal poses are much more common than objects in a side view or a back view. When the training images are sampled randomly, it is much more likely to sample mostly frontal views, which makes our semi-supervised few-shot learning setting even more challenging. The importance of this problem has been recognized in semi-supervised learning for image classification, such that recent works focus entirely on semi-supervised learning from imbalanced data [J]. To avoid further increasing the difficulty of our challenging learning setting, we sample the few labeled images used during training such that the samples are widely distributed in the pose space.
> 2) Related work on semi-supervised learning in image classification also observed that the final model performance is highly affected by the labeled data, particularly in an extreme few-shot setting (Section 4.4. in [K]), where the authors report a performance drop from 78% to 64% by using different random samples of annotated data.
>
> [J] Wei, Chen, et al. "Crest: A class-rebalancing self-training framework for imbalanced semi-supervised learning." Proceedings of the IEEE/CVF Conference on Computer Vision and Pattern Recognition. 2021.
>
> [K] Sohn, K., Berthelot, D., Li, C. L., Zhang, Z., Carlini, N., Cubuk, E. D., ... & Raffel, C. (2020). Fixmatch: Simplifying semi-supervised learning with consistency and confidence. arXiv preprint arXiv:2001.07685.
>
> [L] Oreshkin, B. N., Rodriguez, P., & Lacoste, A. (2018). Tadam: Task dependent adaptive metric for improved few-shot learning. arXiv preprint arXiv:1805.10123.
>
> **Q2: From Table 1, the proposed method does not seem to perform as well as NeMo for 50 training samples. This behavior for larger number of trained samples is not observed for the KITTI datasets, however. So, perhaps it is an artifact of the Pascal3D+ dataset. I would like to hear a further discussion of this from the authors and whether they have analyzed the cause of this observation.**
>
> This observation is indeed an artifact of the Pascal3D+ dataset, which is more challenging than the KITTI dataset. Pascal3D+ has on average 500 (unlabeled) training images per category, and in addition, a larger variability of the 3D object poses. In contrast, KITTI has about 2000 (unlabeled) training images per category and is less variable in the 3D pose than Pascal3D+. On the other hand, the results also demonstrate that our model can leverage the additional unlabelled training data in KITTI very well.
>
> **Q3: I am also curious as to what happens with greater than 50 labeled samples and for 100% labeled training data? Does the proposed approach perform as well as NeMo on both datasets in fully labeled scenarios as well or is it limited in applicability only to less labeled scenarios?**
>
> The kernel component of our approach is pseudo labeling images with matching and bootstrap semi-supervised learning with them. However, for the fully supervised setting, all the images are provided with groundturth labels, which seems unreasonable to conduct our approach on it.
>
> Here we provide the result for when using 100 annotated image, notice that for our approach more annotation can still improve the final pose estimation performance.
>
> | Evaluation Metric   | Pi/6  | Pi/18 | Med  |
> |--------------------------|--------|--------|--------|
> | Num Annos = 7     | 53.8  | 27.0  | 37.5 |
> | Num Annos = 20   | 61.7  | 34.0  | 28.7 |
> | Num Annos = 50   | 65.6  | 39.8  | 24.2 |
> | Num Annos = 100 | 66.8  | 41.7  | 23.6 |
>
> **Q4: Few-shot Viewpoint estimation has been explored previously in the work: Few-Shot Viewpoint Estimation, Tseng et al, BMVC 2019. However it was in a different context of exploring few-shot generalization to previously unseen categories of objects, which is different from this current work. Nevertheless it would be good to cite this closely related prior work.**
>
> Thank you for the suggestion, we will include it in the discussion of our related work.

---

> > ### Comment · Reviewer_9Mv2 · 2021-09-01
> > **Response to Author(s)**
> >
> > I thank the authors for their detailed responses.
> >
> > I am satisfied with the authors' response to Q1. and I encourage them to revise the paper with the discussion of the choice of labeled samples as they promised.
> >
> > With regards to question 3, do the authors have the corresponding results for NeMo with 100 Annotations as well?

---

> > > ### Author Response · Authors · 2021-09-01
> > > **Re: Response to Author**
> > >
> > > **I am satisfied with the authors' response to Q1. and I encourage them to revise the paper with the discussion of the choice of labeled samples as they promised.**
> > >
> > > We will certainly include the results from our discussion in the revision.
> > >
> > > **With regards to question 3, do the authors have the corresponding results for NeMo with 100 Annotations as well?**
> > >
> > > We cannot do this additional experiment before the end of the discussion period since it will need approximately 2-3 days to re-train all models and evaluate their performance. But we will add both our previously reported results from the rebuttal and baseline result for 100 images in the revision.

---

> > > > ### Comment · Reviewer_9Mv2 · 2021-09-01
> > > > **Re: Re: Response to Author**
> > > >
> > > > Thanks for your response. I understand and encourage you to do so in the final version of your paper.

---

> > ### Comment · Reviewer_9Mv2 · 2021-09-01
> > **Final Rating**
> >
> > I am satisfied with the authors' responses to my concerns. I will maintain my original rating and recommend accepting this paper to NeuRIPS.

---

### Official Review · Reviewer_B1pt · 2021-07-16

**Rating:** 4
**Confidence:** 3

**Summary:**

The paper presents a learning based method for 3D pose estimation from single image that works with very few annotated data. Given some exemplar images with 3D pose annotation, the 3D pose of a test image could be estimated by two steps: 1) neural view synthesis and 2) feature matching. In the neural view synthesis step, a 3d feature is generated by unprojecting a pretrained 2d feature map of the image. The 3d feature is then probably "projected" to a set of novel views for feature matching and 3d pose estimation.

**Limitations And Societal Impact:**

Yes.

**Main Review:**

Weakness/questions:

The paper proposed a few shot learning method for 3D pose estimation from 2d images that utilizes the underlying 3D structure. A 2d feature map is unprojected to a spatial grid and this kind of 3D representation is desired to be pose-invariant. However, it is not very clear to me how occlusion reasoning is performed here and how a holistic shape representation is inferred from a single image here. For example, how are the features on the grid are sampled from a 2D view? Is any self-occlusion taken into account when the 2d feature are unprojected to the 3d grid? How about the features at the occluded vertex? Do they take the feature from the object that is in front of them? Also in the neural view synthesis step, how is the rasterization performed? Are all vertex considered to be non-transparent? it is a bit unclear to me for this part and I hope the author could provide more details about this part.

This paper seems to be similar to another relevent work NeMo which is also cited as [44] in this paper. I am wondering if the author could elaborate more on the difference between those two works and the novelty of this work among NeMo?

**Time Spent Reviewing:**

2h

---

> ### Author Response · Authors · 2021-08-10
> **Response to Reviewer B1pt**
>
> We thank the reviewers for their very helpful feedback. In the following, we address each individual question in detail. We ask the reviewers to please follow-up with us in the discussion period in case of any remaining questions.
>
>
> **Potential major misunderstanding of our method.**
> Based on the given summary of our method by Reviewer B1pt, we have the impression that there is a major misunderstanding of our method. In his/her summary, the reviewer says that "given some exemplar images with 3D pose annotation, the 3D pose of a test image could be estimated [...]".  However, we point out that the exemplar **images with 3D pose annotation are only used to pseudo-label unlabelled training images and not the test images** (lines 10-12 in the abstract, lines 211-213).
>
> In particular, the annotated training images are only used in the very first iteration of our EM-type training method to bootstrap the learning process by initializing the model parameters. After the model parameters are initialized, the model trains itself in a self-supervised manner through the proposed neural view synthesis and matching framework. During inference, the trained model is used to estimate the 3D pose of the object in a render-and-compare optimization process, which does not require any annotated data. We ask Reviewer B1pt to please ensure a correct understanding of our approach and also to consider the feedback from the other reviewers, which is generally much more positive.
>
> **Q1: However, it is not very clear to me how occlusion reasoning is performed here and how a holistic shape representation is inferred from a single image here. For example, how are the features on the grid are sampled from a 2D view?**
>
> In general, we differentiate between two types of occlusion: 1) Self-occlusion and 2) partial occlusion by other objects. In our answer to your next question Q2, we explain how self-occlusion is naturally taken into account by the rasterization process. Partial occlusion due to other objects is handled using a robust reconstruction loss as is commonly used in robust statistics [G]. Details on this can be found in other analysis-by-synthesis approaches [H,I]. Intuitively, each position in the feature map is either explained by the object model or an outlier model. In our experiments, the outlier model is a simple Gaussian distribution that is estimated from all training images. Since this is a commonly used approach in related work on occlusion robustness with analysis-by-synthesis approaches, we did not discuss it in detail. Still, we agree that we should include a brief description of the process in the experiments section of the paper.
>
> Concerning the second part of your question, we sample the image features according to the position of the projected mesh vertices.
>
> [G] Huber, P. J. (2004). Robust statistics (Vol. 523). John Wiley & Sons.
>
> [H] Romdhani, S., & Vetter, T. (2003, October). Efficient, robust and accurate fitting of a 3D morphable model. In Computer Vision, IEEE International Conference on (Vol. 2, pp. 59-59). IEEE Computer Society.
>
> [I] Wang, A., Kortylewski, A., & Yuille, A. (2020, September). NeMo: Neural Mesh Models of Contrastive Features for Robust 3D Pose Estimation. In International Conference on Learning Representations.
>
> **Q2: Is any self-occlusion taken into account when the 2d feature are unprojected to the 3d grid? How about the features at the occluded vertex? Do they take the feature from the object that is in front of them? Also in the neural view synthesis step, how is the rasterization performed? Are all vertex considered to be non-transparent? it is a bit unclear to me for this part and I hope the author could provide more details about this part.**
>
> Self-occlusion is taken into account through the rasterization process (as mentioned in line 159). We follow the standard rasterization procedure defined in the pytroch3D library, which is commonly used in computer graphics rendering pipelines. In particular, the feature values at a pixel in the output image are generated using the mesh triangle that is closest to the camera on the viewing ray from the camera to the pixel. In this way, self-occluded triangles do not influence the pixel value and are ignored. In this way, only the features at the visible vertices are updated using Equation 6. All vertices are considered to be non-transparent.
>
> **Q3: This paper seems to be similar to another relevent work NeMo which is also cited as [44] in this paper. I am wondering if the author could elaborate more on the difference between those two works and the novelty of this work among NeMo?**
>
> NeMo is a fully supervised approach to 3D pose estimation and hence requires a large amount of annotated data during training. Obtaining annotated data is time-consuming and expensive. Our work introduces a semi-supervised few-shot learning approach that performs 3D pose estimation from minimal human supervision using very few annotated examples and a collection of unlabelled images. In particular, we introduce an EM-type training strategy in which we transfer the 3D pose annotation from the few labeled examples to the unlabelled data. Our experiments demonstrate that our proposed training strategy enables highly efficient learning of 3D pose, particularly in extreme few shot scenarios, outperforming NeMo and other baselines by a wide margin. Our work shows that 3D object representations can be learned very efficiently from a collection of unlabelled data by leveraging prior knowledge about the 3D nature of objects.  Our learned 3D representations also achieve strong robustness in out-of-distribution scenarios that involve occlusion, despite being learned using minimal supervision. On a conceptual level, our work demonstrates the benefits of 3D representations in terms of learning efficiency and robustness, and we hope that this will inspire future work in other computer vision areas such as image classification or object detection.

---

> > ### Comment · Reviewer_B1pt · 2021-08-21
> > **Response to the author(s)**
> >
> > Thanks for the author(s) for those responses. After viewing those answers, I think I could be convinced that few-shot learning and label propagation could be novelties of this work and I am fine with raising my rating. However, the rendering and geometric level feature learning part is still not very clear to me. For example, on handling the self-occlusion part, the author(s) said that all vertices are considered non-transparent which confused me a bit. In a way that the occupancy or transparency of the cuboid grid is not optimized, the worst case where the grid resolution is infinitely large, only the features on the outer surface is rasterized and all features inside would be blocked. And a small change in viewpoint would probably also cause some of the inner grid feature visible through the "hole" on the surface. So in that way, I am not saying that why the self-occlusion is handled in a geometrically consistent way as the occupancy of the grid is not matching the real occupancy of certain objects. Setting aside all those doubts, I am fine with raising my scores.

---

> > > ### Author Response · Authors · 2021-08-21
> > > **Response to Reviewer B1pt**
> > >
> > > We thank Reviewer B1pt for giving us the opportunity to clarify open questions.
> > >
> > > **After viewing those answers, I think I could be convinced that few-shot learning and label propagation could be novelties of this work and I am fine with raising my rating.**
> > >
> > > We also thank the reviewer for raising the rating.
> > >
> > > **Q1: The rendering […] is still not very clear to me. […] The worst case where the grid resolution is infinitely large, only the features on the outer surface is rasterized and all features inside would be blocked. And a small change in viewpoint would probably also cause some of the inner grid feature visible through the "hole" on the surface**
> > >
> > > We want to clarify that our mesh cuboid representation is not affected by "holes" during the rasterization. Our mesh has no holes, it is one closed surface. We think the reviewer's question is caused by an unclarity about the classic rasterization process (which is also mentioned by the reviewer in his comment). Therefore, we want to explain the classic rasterization process in more detail. In computer graphics, a mesh is a set of 3D vertices that are connected with edges into triangle faces (see Figure 1 below). When the mesh is projected into the image plane, some triangles will occlude other triangles, and the rasterization process handles this self-occlusion by finding the front-most triangle for every pixel (this process is illustrated in Figure 1, where only the triangles of the dolphin mesh are rendered that are visible to the viewer).
> > >
> > > Concerning the problem which the reviewer refers to as "holes" – During the rasterization, some pixels will not be covered by the projected vertices (see e.g., the red pixel in the schematic illustration in Figure 2). Filling these "holes" is exactly the task of the standard rasterization process. Specifically, the feature vector at the red pixel is generated by interpolating the features at the vertices of the triangle that covers this pixel. In our case, the pytroch3D library implements a standard linear interpolation of the vertex features using the weighted distance of the pixel to the individual vertices. As our mesh cuboid is a closed surface with no holes, the rasterization process will generate a feature vector for every pixel that is in-between the projected vertices of a triangle. Note that this process can handle arbitrary grid resolutions.
> > >
> > > We hope that our explanation can clarify the reviewer’s concerns. We encourage the reviewer to please reach out to us in case of any further concerns.
> > >
> > > Figure 1: https://drive.google.com/file/d/1woRrKOALPPgMazK1ttn7LvbJGe4mehUS/view?usp=sharing
> > >
> > > Visualization of a rendered mesh. We note that the self-occlusion is handled by the rasterization process, and the triangles on the backside of the dolphin are not rendered.
> > >
> > > Figure 2: https://drive.google.com/file/d/1KXF1tp7CmzXKsQ4hTQD3K3Xy3nLQFXPK/view?usp=sharing
> > >
> > > Visualization of the pixel interpolation in the rasterization process. We note that the pixel marked in red is not covered by the three vertices of the triangle mesh. Instead, the features at the pixel position will be generated by interpolating the features of the three triangle vertices.

---

> > > > ### Comment · Reviewer_B1pt · 2021-08-21
> > > > **Responses to the author(s)**
> > > >
> > > > Thanks for those responses. But I am still not clear that about the mesh cuboid part. Please correct me if I am wrong. If all the surface are non-transparent in a mesh cuboid, I assume only the OUTER surface feature will be rendered. And if I am understanding it correctly, the geometry of the mesh cuboid is not matching the geometry of the object that appeared in the image. So in that case, there will be no guarantee on the geometric consistency between the feature map under current viewpoint and the feature map under the novel viewpoint. For example the feature of the front windshield might be rendered into the area that is in front of the car front windshield under certain view, due to the mismatching geometry between the car and a cuboid. And in section 3.2 equation (3), the similarity between two feature map is directly computed by a deterministic function which is cosine distance. I also wonder how those spatially misaligned features could be used to compute a pixel-wise cosine distance without any warping model to correct the misalignment? In the case of cars, their shapes are relatively convex. I am also wondering how non-convex shape objects are handled?

---

> > > > > ### Author Response · Authors · 2021-08-24
> > > > > **Responses to Reviewer B1pt**
> > > > >
> > > > > Yes, the geometry of the mesh only very roughly approximates the shape of the object. But we note that our approach already achieves a very good performance considering the minimal amount of supervision used. We think that our model achieves a high performance despite the mismatch between the cuboid and the object geometry because:
> > > > >
> > > > > 1. Our spatial matching is conducted on the level of neural network features. Compared to a pixel-level matching, the matching of neural features is significantly less sensitive to a misalignment in terms of the object geometry. The reason is that the neural network features are typically invariant to small variations in nuisance variables such as shape deformations or changes in the object texture and illumination (as mentioned in l.57-59). This invariance is further enhanced as our feature extractor is trained using the matched features after each Neural Synthesis and Matching iteration.
> > > > >
> > > > > 2. We extract features at the high convolution layer of ResNet50, which have a relatively large receptive field. In combination with our contrastive loss (Equation 4 and 5), the feature vectors that are "spatially misaligned" with the object geometry can still learn to focus on the "aligned" object parts because they are consistent in the training data, whereas the background context will vary (e.g., a feature vector that is located above the car can learn to focus on the roof of the car). Intuitively, the contrastive training strategy can therefore account for the mismatch in the geometry of the cuboid and the object (note that some objects of the Pascal3D+ dataset are non-convex, e.g., the airplane or boat classes).
> > > > >
> > > > > Our motivation for using a cuboid to represent the object geometry is that we conduct category-level pose estimation, i.e., we learn one model that can perform pose estimation for many different subtypes of an object class, with minimal human supervision. In contrast, generating a detailed mesh model for every subtype requires expensive and time-consuming annotation, and learning these meshes in our semi-supervised few-shot setting will be very difficult and beyond the scope of this paper.
> > > > >
> > > > > The question regarding the learning of a deformable mesh model (warping the mesh in 3D space) was also askes by Reviewer kdyM Q1, and we provide our answer in the following. We agree that deforming the mesh representation is a very interesting research direction. However, adding additional degrees of freedom to the mesh would make the learning task even more challenging because it becomes even more under-constrained due to the minimal supervision we use in our work. Learning how to deform the mesh geometry would require additional constraints, such as estimating the object's segmentation mask. We think that this is theoretically possible, but it requires significant extensions of the presented framework that go beyond the scope of this paper.

---

### Official Review · Reviewer_h5BS · 2021-07-16

**Rating:** 7
**Confidence:** 4

**Summary:**

This paper studies the problem of estimating the pose of an object, given an image of it. The proposed method aims to use as few labeled samples as possible, and generate pseudo-labels for a large collection of unlabelled data during training. The method works in an EM-fashion in which two stages (neural view synthesis and matching) are optimized by alternating.

In the first stage, the method extracts a feature map from the input annotated image and shapes it as a mesh cuboid where each vertex is assigned a feature vector depending on its position. The mesh is then rotated at regular angle intervals and rasterized into the image plane. In the matching part, features from unlabeled images are extracted and matched with rotated view rasterizations using a cosine similarity metric.

The experimental evaluations show that the proposed method is able to learn the pose prediction task from as few as 7 annotated images during training. The method mostly outperforms the baselines in comparisons against them.


**Ethical Concerns:**

The paper does not present any ethical concerns to the best of my knowledge.

**Limitations And Societal Impact:**

The limitations of this work, along with potential societal impacts are discussed in the conclusion. The inability of trivial generalization to articulated objects such as humans is listed as a limitation. I believe that there are others as well though, such as the existence of a large error margin (40%) for pose estimation even when the model is evaluated at 30 degree thresholds. I encourage the authors to detail the limitations section of the paper.


**Main Review:**

The proposed method is novel in the way that it generates 3D pose pseudo-labels for large amounts of unlabeled data which is difficult to annotate in practice. I believe that the method can be useful in future research in this area.

Overall, I think the paper is organized well. There are a few typos, some of which are included at the end of this review. My questions and comments are listed below.

- The term neural view synthesis sounds somewhat misleading in my opinion, since the proposed method does not synthesize and render RGB images in the way the literature on view synthesis does. I would suggest clarifying what is meant by view synthesis and adding how it is different from the literature.

- How does the method ensure that the extracted features from the image have meaningful 3D information? What does it mean to rotate the features extracted by a network pre-trained on ImageNet?

- How does the method avoid trivial solutions of assigning the nearest neighbor’s label as the pseudo-label? What is the intuition behind preventing such mode collapses? For instance, if there are images with labels of 30 and 45 degrees azimuth in the annotated dataset, how does the method infer an angle of 35 degrees?

- Similarly, how is the model able to differentiate between the pose angles from the front and back views, or from left and right side views? Are there any failure modes involving such examples?

- Do the results in the tables average over all three output angles (azimuth, elevation and in-plane rotations)? How does the method perform in these individual terms?

- Is ∆θ only along the azimuth axis or does it also include the elevation and in-plane rotations?

- Can you detail the optimization at test time more? Does the method use additional labeled data at this stage? What is the computation time of the pose optimization at inference time?

- What is the procedure to select the few images with labels? Does it need to cover the viewing hemisphere? Any discussion would be helpful to include in my opinion.

- Does the approach of averaging features from multiple annotated images have any drawbacks? Can having a large number of labeled data (> 100) degrade the performance due to features averaging out each other?

- What is the curriculum strategy for starting with small ∆θ and increasing it while training?

- Why are the angle errors increased for the case of ∆θ = 0, after the semi-supervised training in Figure 2?


Minor comments:

- Why is the feature cuboid named a mesh rather than a voxel grid? Since the mesh is a grid, and the edges and faces are not used, I believe calling it a voxel grid is more accurate.
- What do the features projected to the image plane look like? I would suggest adding those generated feature images to the paper.
- Please include the standard deviation of the results in the tables as well, for a more complete evaluation.
- Typos: line 53: the the; line 63: for for; line 65: annotated -> annotate, line 78: between between, line 161: wholes -> holes

Update: after my concerns were clarified in the response, I raised my score to accept.


**Time Spent Reviewing:**

4

---

> ### Author Response · Authors · 2021-08-10
> **Response to Reviewer h5BS**
>
> We thank the reviewers for their very helpful feedback. In the following, we address each individual question in detail. We ask the reviewers to please follow-up with us in the discussion period in case of any remaining questions.
>
> **A potential misunderstanding of our approach.**
> Based on Reviewer h5BS's summary of our method and some of his/her questions, we think there might be a misconception of some of the technical details of our method. In particular, the annotated training images are only used in the first iteration of our EM-type training method to bootstrap the learning process by initializing the model parameters. After the model parameters are initialized, the model trains itself from the unlabeled training data in a self-supervised manner through the proposed neural view synthesis and matching framework. During inference, the trained model is used to estimate the 3D pose of the object in a render-and-compare optimization process, which does not require any annotated data.
>
> **Q1: The term neural view synthesis sounds somewhat misleading in my opinion, since the proposed method does not synthesize and render RGB images in the way the literature on view synthesis does. I would suggest clarifying what is meant by view synthesis and adding how it is different from the literature.**
>
> (This question is related to Q3 of Reviewer kdyM)
>
> We will discuss the difference between our approach of rendering on the feature level to related work on view synthesis with RGB images in the introduction and related work in more detail. One important difference is that related works for neural view synthesis do not focus on 3D pose estimation but rather image synthesis. However, performing inverse rendering with models that synthesize on the level of RGB pixels is challenging because the pixel level reconstruction loss is very difficult to optimize, e.g., see [A]. In contrast, we synthesize on the level of neural network features, which are trained to be invariant to 3D pose variations and intra-class variabilities such as changes in the texture and appearance. Our feature-level reconstruction loss can easily be optimized with a simple gradient descent optimization, because we train the features to be invariant to object details that are irrelevant for the 3D pose estimation task.
>
> [A] Schönborn, S., Egger, B., Morel-Forster, A., & Vetter, T. (2017). Markov chain monte carlo for automated face image analysis. International Journal of Computer Vision, 123(2), 160-183.
>
> **Q2: How does the method ensure that the extracted features from the image have meaningful 3D information? What does it mean to rotate the features extracted by a network pre-trained on ImageNet?**
>
> We do not rotate the features. We sample the features at the mesh vertices and rotate the mesh. This is an important difference since rotating the mesh only changes the spatial position of the features but not their values. This process is similar to the rendering of RGB images with a computer graphics model. The RGB values at the mesh vertices are not rotated; what is rotated is the mesh, which only changes the position of the vertex (note that the main difference of our method to the rendering of RGB images is that we only rotate and rasterize and do not model illumination, because our features are trained to be invariant to changes in the appearance).
>
> It is a bit unclear to us what the reviewer means by "meaningful 3D information". We train the features to be invariant to changes in the 3D object pose with a contrastive loss. This means that the features at a particular mesh vertex should be as similar as possible to each other, independent of the 3D pose of the mesh (see Equation 4, lines 188-193). Features of different vertices are trained to be as distinct from each other as possible (Equation 5) to enable an accurate 3D pose estimation.
>
> **Q3: How does the method avoid trivial solutions of assigning the nearest neighbor's label as the pseudo-label? What is the intuition behind preventing such mode collapses?**
>
> This question is unclear since our matching approach assigns the pseudo-labels to the nearest neighbor's w.r.t. the similarity measure defined in Equation 3 by comparing images on the feature level. Our goal is to train the feature extractor such that this similarity is indeed a good measure for the distance in the 3D pose between two images.
>
> In the following, we assume that the reviewer's question refers to the early stages of the training when the feature extractor is not yet fully trained. In this case, the question is related to Q2 from Reviewer kdyM. An important mechanism that enables our proposed approach to predict pseudo-labels reliably is that we gradually extend the 3D pose of the pseudo-labeled data during training, as described in lines 173-181. In addition, we found that the following two strategies are very useful to predict pseudo-labels reliably:
> 1) When synthesizing novel views, we limit the difference ∆θ of the unseen poses to the already learned poses to be small (increasing ∆θ by 10 degrees). This forces the model to explore unseen poses slowly, which makes the EM-type learning more stable.
> 2) To prevent the algorithm from confusing similar-looking poses (front/back and left/right facing objects), we always sample a set of 4 azimuth poses during training which is front-back and left-right symmetrical (i.e., φ, 180 - φ, 180 + φ, 360 - φ). This strategy introduces a competition between similar-looking poses, which resolves the confusion of symmetrical poses along the azimuth angle. We will update our paper to include these implementation details and include a description in the comments of our code upon the public release.
>
> These mechanisms cannot entirely avoid incorrect pseudo-labels (e.g., as shown in the 4th row of the 1st column in Figure 2). Still, if the overall number of wrong predictions is sufficiently low, our model generally converges to a very good solution during the EM-type learning process because the parameter updates are performed in a robust manner:
> 1) By using a moving average update (Equation 7) during learning, which makes the parameter estimation process robust even when there are some errors in the pseudo labeling.
> 2) By using a relatively large batch size of 80 training images. If some of those training images are labeled incorrectly, they only have a limited influence on the result.
>
> **Q4: For instance, if there are images with labels of 30 and 45 degrees azimuth in the annotated dataset, how does the method infer an angle of 35 degrees?**
>
> It is unclear if the reviewer refers to the training or inference stage in this question. We think this is due to a misunderstanding of important parts of our work, and we discussed this at the beginning of our response in more detail.
>
> We assume that the reviewer is referring to the training stage of our algorithm since the labeled images are not used during testing. During training, we synthesize novel views at discrete poses and match these to the unlabeled data to create pseudo-labels, which are in turn used to update the model parameters. Our experiments show that such a discrete pose sampling at training time is sufficient to learn a feature representation that is invariant to changes in the 3D pose. At test time, we estimate a continuous 3D pose using a render-and-compare optimization process (see our answer to Q8 for more details on the inference process).
>
> **Q5: Similarly, how is the model able to differentiate between the pose angles from the front and back views, or from left and right side views? Are there any failure modes involving such examples?**
>
> The front-back and left-right confusion are among the main challenges in the semi-supervised few-shot learning of 3D pose. We introduced a technical solution into the learning process that largely resolves this problem (we also describe this in our answer to your question Q3): To prevent the algorithm from confusing similar-looking poses (front/back and left/right facing objects), we always sample a set of 4 azimuth poses during training which is front-back and left-right symmetrical (i.e., φ, 180 - φ, 180 + φ, 360 - φ). This strategy introduces a competition between similar-looking poses, which resolves the confusion of symmetrical poses along the azimuth angle. We will update our paper to include these implementation details and include a description in the comments of our code upon the public release.
>
> **Q6: Is ∆θ only along the azimuth axis or does it also include the elevation and in-plane rotations? & What is the curriculum strategy for starting with small ∆θ and increasing it while training?**
>
> We sample the pose along all three pose angles (azimuth, in-plane, and elevation). We uniformly sample a fixed step size by increasing ∆θ in 10 degree steps along all three pose angles. Starting from the initially labeled poses, we render novel views by making one step ∆θ in all three pose angles. The synthesized views are used to pseudo-label some of the unlabelled data and to subsequently update the model parameters. In the next learning iteration, we further increase the pose of the synthesized views by 10 degrees to obtain pseudo-labeled data and to update the model parameters. We continue this process until the full 3D pose space is explored. This gradual exploration of the pose space stabilizes the EM-type learning. This process will also be documented upon the public release of our code and trained models.

---

> ### Author Response · Authors · 2021-08-10
> **Continue: Response to Reviewer h5BS**
>
> **Q7: Do the results in the tables average over all three output angles (azimuth, elevation and in-plane rotations)? How does the method perform in these individual terms?**
>
> As we described in Section 4.1. (lines 233-237), we follow the standard evaluation protocol, which is commonly used in related work on 3D pose estimation [C,D]. In particular, we evaluate the accuracy of all three pose angles by computing the geodesic distance between the ground truth rotation matrix and the predicted rotation matrix.
>
> [C] Tulsiani, S., & Malik, J. (2015). Viewpoints and keypoints. In Proceedings of the IEEE Conference on Computer Vision and Pattern Recognition (pp. 1510-1519).
> [D] Zhou, X., Karpur, A., Luo, L., & Huang, Q. (2018). Starmap for category-agnostic keypoint and viewpoint estimation. In Proceedings of the European Conference on Computer Vision (ECCV) (pp. 318-334).
>
> **Q8: Can you detail the optimization at test time more? Does the method use additional labeled data at this stage? What is the computation time of the pose optimization at inference time?**
>
> (This question is related to Q3 of Reviewer kdyM)
>
> We do not use any groundturth labels at test time. The computation time is on average less than 3s per image on an Nvidia Titan RTX GPU.
>
> During inference, we first use the trained model to synthesize a set of predefined initial starting poses (in practice, we use ten starting poses that are distributed across the pose space). The starting pose with the highest similarity to the feature map of the test image (Eq. 3) provides an initial coarse estimate of the object's 3D pose in the test image. Following previous works on analysis-by-synthesis [E, F], we continue to refine the coarse pose prediction through a render-and-compare optimization process. Specifically, starting from the initial pose estimate, we optimize the 3D pose by computing the gradient w.r.t. the similarity measure (Eq.3) using the Adam optimizer. We run the gradient-based optimization for 200 steps.
> Note that the initialization process during inference is similar to the neural view synthesis and matching process during training. However, during inference, we find the most similar synthesized feature map for a given test image. Whereas during training, we find a set of most similar unlabelled training images for each synthesized view.
>
> [E] Blanz, V., & Vetter, T. (1999, July). A morphable model for the synthesis of 3D faces. In Proceedings of the 26th annual conference on Computer graphics and interactive techniques (pp. 187-194).
> [F] Wang, A., Kortylewski, A., & Yuille, A. (2021). NeMo: Neural Mesh Models of Contrastive Features for Robust 3D Pose Estimation. arXiv preprint arXiv:2101.12378.
>
> **Q9: What is the procedure to select the few images with labels? Does it need to cover the viewing hemisphere? Any discussion would be helpful to include in my opinion.**
>
> In the experimental setting where we use 7 labeled images, the images are manually selected to be roughly uniformly distributed in the pose space. To illustrate this, we show the images used to train the car model in the following anonymous google drive link (images are cropped for visualization): https://drive.google.com/file/d/1glLmue-g3AxilWx2IwA6cdN-7wn7IVws/view?usp=sharing
> In our answer to Q1 of Reviewer 9Mv2, we provide additional results with randomly selected starting images.
>
> **Q10: Does the approach of averaging features from multiple annotated images have any drawbacks? Can having a large number of labeled data (> 100) degrade the performance due to features averaging out each other?**
>
> We think averaging features will not have significant drawback, since once our semi-supervised training will make the feature tightly clustered that all feature representation will become similar to the mean feature. On the other hand, it is not computationally feasible to retain features from all retrieved images and synthesis them separately. Also as we describe in our answer to R2 Q3, averaging features can make our training more robust to errors during retrieve.
>
> Here we provide the result for when using 100 annotated image, notice that for our approach more annotation can still improve the final pose estimation performance.
>
> | Evaluation Metric   | Pi/6  | Pi/18 | Med  |
> |--------------------------|--------|--------|--------|
> | Num Annos = 7     | 53.8  | 27.0  | 37.5 |
> | Num Annos = 20   | 61.7  | 34.0  | 28.7 |
> | Num Annos = 50   | 65.6  | 39.8  | 24.2 |
> | Num Annos = 100 | 66.8  | 41.7  | 23.6 |
>
>
> **Q11: Why are the angle errors increased for the case of ∆θ = 0, after the semi-supervised training in Figure 2?**
>
> We think this is due to a trade-off between specificity and generalization ability of the features. In particular, if the features are very specific, they match very accurately to objects in the same 3D pose. On the other hand, if the features generalize well, they can match a wide range of different 3D poses. Figure 4 shows that our semi-supervised training will significantly increase the generalization ability of the feature extractor, which can be used to match a wide range of poses. However, this comes at the cost of a small reduction of the specificity of the features.
>
> **Q12: Why is the feature cuboid named a mesh rather than a voxel grid? Since the mesh is a grid, and the edges and faces are not used, I believe calling it a voxel grid is more accurate.**
>
> Voxel grids are a dense representation that also represents the inside of the object. Voxel representations are, for example, very frequently used in medical imaging (e.g., to represent CT or X-ray scans). In our work, we only want to represent the surface of the object and not the inside. Therefore, a mesh-based representation is more suitable in our approach.
>
> **Q13: What do the features projected to the image plane look like? I would suggest adding those generated feature images to the paper.**
>
> We have tried visualizing the features using PCA, but we found that this is infeasible due to the large size of the feature maps.
>
> **Q14: Please include the standard deviation of the results in the tables as well, for a more complete evaluation.**
>
> Thank you for the suggestion, we will add them.
>
> **Q15: Typos: line 53: the the; line 63: for for; line 65: annotated -> annotate, line 78: between between, line 161: wholes -> holes**
> Thank you for pointing out these typos, we will fix them.

---

> > ### Comment · Reviewer_h5BS · 2021-08-30
> > **Response to the authors**
> >
> > Thank you for the detailed response. I appreciate that the authors responded and clarified almost all of the concerns I had in the initial review. After reading the rest of the reviews and the responses to them, I am more in favor of the acceptance of this paper and I raised my score.
> >
> > I believe that adding the discussion in the response regarding the algorithm’s performance in the training stage would be helpful. I also encourage the authors to include the additional experiments they made in the next version of the paper.

---

### Official Review · Reviewer_kdyM · 2021-07-19

**Rating:** 7
**Confidence:** 4

**Summary:**

This paper presents a method for 3D pose estimation of PACAL3D objects (6 vehicle categories). The method bootstraps from 7-50 annotations per category. The idea is: given the initial annotations, finetune a Resnet50 and a featurized cuboid mesh with contrastive losses that bring the 2D and 3D features close to each other, and make them discriminative for position/viewpoint. Then, compute 2D features for the entire dataset, and rotate the 3D featuremap a few degrees in each direction, and compute match scores for all pairs. The confident matches are then used for subsequent re-optimization of the 2D and 3D features, and this process repeats ~100 times, gradually increasing the number of rotation bins covered by the 3D featuremap. Results are quite good against non-bootstrapping baselines, trained just with the 7-50 annotations.


**Ethical Concerns:**

No concerns.

**Limitations And Societal Impact:**

No concerns.

**Main Review:**

I like this paper. Pose estimation is a hard problem, and this boot-strapping/EM solution combines insights from pre-trained ImageNet feature-learning and structured 3D representations in a clever way, to gradually cover the full set of viewpoints/angles of the objects. The approach is a little constrained, in terms of assuming perfect detections and known scale/distance and centroid, but maybe this is OK.

I was surprised that the approach does not also involve some optimization of the 3D mesh of the category, and instead keeps it as a cuboid. Have the authors tried optimizing this, or perhaps optimizing occupancy inside the grid, and using occupancy as part of the rendering?

What about the drift? EM/bootstrapping methods like this inevitably drift (and also sometimes recover). I think it is important to show pose estimation errors over EM rounds, and maybe to visualize some "hard" examples for which the the wrong pose has a better match score than the right pose.

How does the pose analysis-by-synthesis work exactly? Is it SGD on the three angles? Does this require multiple initializations, or does it work even if it starts 180deg off?

I understand the idea to compare against baselines under the same semi-supervised setting, but I think it is important to also report those models' fully-supervised results, to put things into perspective.

It would be interesting (but maybe not essential) to compare against "Image GANs meet differentiable rendering for inverse graphics and interpretable 3d neural rendering" (ICLR 2020), which I think can also be interpreted as an EM approach: an image generator is used to optimize a 3D inverse graphics net, and the inverse graphics net is used to optimize the latent space of the image generator. I do not think they evaluated on pose estimation, but they could have.

It would be nice to visualize the 2D and 3D features, with PCA.

Some typos:
- Groundturth -> Groundtruth
- wholes -> holes

**Time Spent Reviewing:**

1.5

---

> ### Author Response · Authors · 2021-08-10
> **Response to Reviewer kdyM**
>
> We thank the reviewers for their very helpful feedback. In the following, we address each individual question in detail. We ask the reviewers to please follow-up with us in the discussion period in case of any remaining questions.
>
> **Q1: I was surprised that the approach does not also involve some optimization of the 3D mesh of the category, and instead keeps it as a cuboid. Have the authors tried optimizing this, or perhaps optimizing occupancy inside the grid, and using occupancy as part of the rendering?**
>
> This is indeed a very interesting research direction, and we are exploring this in our future work. However, adding additional degrees of freedom to the mesh would make the learning task even more challenging because it becomes even more under-constrained due to the minimal supervision we use in our work.  Learning how to deform the mesh geometry would require additional constraints, such as estimating the object's segmentation mask. We think that this is theoretically possible, but it requires significant extensions of the presented framework that go beyond the scope of this paper.
>
> **Q2: What about the drift? EM/bootstrapping methods like this inevitably drift (and also sometimes recover). I think it is important to show pose estimation errors over EM rounds, and maybe to visualize some "hard" examples for which the the wrong pose has a better match score than the right pose.**
>
> Drift is indeed a common problem in EM-type optimization methods. We observe empirically that our EM-type optimization process generally converges to a very good solution. The most important properties that make the optimization process robust are:
> 1) We use a moving average update (Equation 7) during learning, which makes the parameter estimation process robust even when there are some errors in the pseudo labeling.
> 2) We use a relatively large batch size of 80 training images. If some of those training images are labeled incorrectly, they only have a limited influence on the result.
>
> When combined, 1) + 2) make the learning process very stable if the model does not make many pseudo-labeling errors.
>
> To reduce the number of wrong pseudo-labels, we gradually bootstrap the model, as described in lines 173-181. In particular, we found that the following strategies are beneficial to predict accurate pseudo-labels:
> 1. When synthesizing novel views, we limit the difference of the unseen poses to the already learned poses  ∆θ to be small (increasing ∆θ in 10 degree steps). This makes the exploration speed of unseen poses slow but more stable.
> 2. To prevent the algorithm from confusing similar-looking poses (front/back and left/right facing objects), we always sample a set of 4 azimuth poses during training which are front-rare and left-right symmetrical (i.e., φ, 180 - φ, 180 + φ, 360 - φ). This strategy introduces a competition between similar-looking poses, which resolves the confusion of symmetrical poses along the azimuth angle. We will update our paper to include these implementation details and include a description in the comments of our code upon the public release.
>
> The pose estimation errors in different epochs of the EM-type learning process are visualized in Figure 4b.
>
> **Q3: How does the pose analysis-by-synthesis work exactly? Is it SGD on the three angles? Does this require multiple initializations, or does it work even if it starts 180deg off?**
>
> During inference, we first use the trained model to synthesize a set of predefined initial starting poses (in practice, we use ten starting poses that are distributed across the pose space). The starting pose with the highest similarity to the feature map of the test image (Eq. 3) provides an initial coarse estimate of the object's 3D pose in the test image.
> Following previous works on analysis-by-synthesis [1,2], we continue to refine the coarse pose prediction through a render-and-compare optimization process.  Specifically, starting from the initial pose estimate, we optimize the 3D pose by computing the gradient w.r.t. the similarity measure (Eq.3) using the Adam optimizer. We run the gradient-based optimization for 200 steps.
>
> Note that the initialization process during inference is similar to the neural view synthesis and matching process during training. However, during inference, we find the most similar synthesized view for a given test image. Whereas during training, we find a set of most similar unlabelled training images for each synthesized view.
>
> [1] Blanz, V., & Vetter, T. (1999, July). A morphable model for the synthesis of 3D faces. In Proceedings of the 26th annual conference on Computer graphics and interactive techniques (pp. 187-194).
> [2] Wang, A., Kortylewski, A., & Yuille, A. (2021). NeMo: Neural Mesh Models of Contrastive Features for Robust 3D Pose Estimation. arXiv preprint arXiv:2101.12378.
>
>
>
>
> **Q4: I understand the idea to compare against baselines under the same semi-supervised setting, but I think it is important to also report those models' fully-supervised results, to put things into perspective.**
>
> We thank the reviewer for the suggestion and will add these baselines to our paper. We include the results below:
>
> | Fully Supervised | Pi/6 | Pi/18 | Med |
> |------------------|------|-------|-----|
> | Starmap          |90.5  | 63.9  | 8.2 |
> | NeMo             |89.3  | 66.7  | 7.7 |
>
> **Q5: It would be interesting (but maybe not essential) to compare against "Image GANs meet differentiable rendering for inverse graphics and interpretable 3d neural rendering" (ICLR 2020), which I think can also be interpreted as an EM approach: an image generator is used to optimize a 3D inverse graphics net, and the inverse graphics net is used to optimize the latent space of the image generator. I do not think they evaluated on pose estimation, but they could have.**
>
> We agree that the suggested ICLR paper is very interesting and related. However, there are significant differences. The paper learns a neural rendering model of RGB images, and it is not applied to 3D pose estimation (as mentioned by the reviewer). It is unclear if their model would perform well at 3D pose estimation as inverse rendering with models that synthesize on the level of RGB pixels is difficult because the pixel-level reconstruction loss is very difficult to optimize, e.g., see [A]. In contrast, we synthesize on the level of neural network features, which are trained to be invariant to 3D pose variations and intra-class variabilities such as changes in the texture and appearance.  Our feature-level reconstruction loss can be easily optimized with a simple gradient descent optimization, because we train the features to be invariant to object details that are irrelevant for the 3D pose estimation task.
>
> [A] Schonborn, S., Egger, B., Morel-Forster, A.,  Vetter, T. (2017).  Markov chain monte carlo for automated face image analysis.  International Journal of Computer Vision, 123(2), 160-183
>
> **Q6: It would be nice to visualize the 2D and 3D features, with PCA.**
>
> This is indeed a very interesting thought and we have considered visualizing the feature representations, but we were unable to compute the PCA components because the feature maps in our model are of size 16 x 42 x 1024, which makes it infeasible due to the very large memory consumption.
>
> **Q7:  Some typos**
> Thank you for pointing out the typos, we will fix these and proof-read the manuscript carefully again.

---

> > ### Comment · Reviewer_kdyM · 2021-09-01
> > **Thanks**
> >
> > Thank you for the detailed responses. Everything makes sense except for the PCA comment. The idea of the visualization is to compress the channel dimension down to 3 principal components, and map those to RGB; you can use a random subset of vectors when computing the principal components if you have memory issues with the full tensor, and the visualization will likely come out the same.
> >
> > One more small point. The response says:
> > > Learning how to deform the mesh geometry would require additional constraints, such as estimating the object's segmentation mask
> >
> > This is not necessarily true. Learning the mesh/occupancy introduces new free variables, but if the EM is very stable, it may work even without additional constraints. It might be fun to try.

---

> > > ### Author Response · Authors · 2021-09-01
> > > **Re: Thanks**
> > >
> > > **The idea of the visualization is to compress the channel dimension down to 3 principal components, and map those to RGB; you can use a random subset of vectors when computing the principal components if you have memory issues with the full tensor, and the visualization will likely come out the same.**
> > >
> > > We thank reviewer kdyM for the suggestion on the PCA visualization, we had initially misunderstood this, but the reviewer's additional comment clarified the idea, and we were able to implement the suggested diagnostic experiment now. In the following Figure, we show the result of the PCA visualization:
> > > https://drive.google.com/file/d/1NEtNjgcyOrDQzCNMj1z2tLt94Fzfki0a/view?usp=sharing
> > >
> > > We followed the suggestion of the reviewer and conduct PCA on the feature vectors at each mesh vertex Σ. We reduce their dimensionality to 3 and encode the PCA embedding as an RGB value. Finally, we render the mesh and RGB values in a given pose into an image. The visualization shows that features which are spatially close are more similar to each other and most features on the mesh have a different color embedding. This is exactly what the contrastive loss in Equations 4 and 5 encourages. However, we note that the first three PCA components only account for about 14.5% of the variation in the data (and hence some features that are very different in the original space might share a similar color in this visualization). Nevertheless, this is an interesting diagnostic experiment.
> > >
> > > Concerning the experiment with the occupancy, we agree that it would be fun to try, but we will not be able to finish it in time before the end of the discussion period.

---

### Decision · Program_Chairs · 2021-09-27

**Decision:**

Accept (Poster)

**Comment:**

This paper addresses semi-supervised 3D viewpoint estimation by lifting 2D CNN features into object-centric feature cuboids, which are rotated and rendered to generate features maps of alternative views, while also updates in a contrastive manner exploiting labelled and pseudo-labelled examples in an iterative manner. All reviewers agree on the novelty of the approach pursued in the paper. The rebuttal submitted by the authors clarified many concerns of the reviewers regarding handling of  occlusions and self-occlusions, scalability of the method with respect to the number of annotated examples, evaluation metric and thresholds used.